# Effects of Prior Infection with SARS-CoV-2 on B Cell Receptor Repertoire Response during Vaccination

**DOI:** 10.3390/vaccines10091477

**Published:** 2022-09-06

**Authors:** Elizabeth R. Fraley, Santosh Khanal, Stephen H. Pierce, Cas A. LeMaster, Rebecca McLennan, Tomi Pastinen, Todd Bradley

**Affiliations:** 1Genomic Medicine Center, Children’s Mercy Research Institute, Children’s Mercy Kansas City, Kansas City, MO 64108, USA; 2Department of Pathology and Laboratory Medicine, University of Kansas Medical Center, Kansas City, KS 66160, USA; 3Department of Pediatrics, University of Missouri-Kansas City School of Medicine, Kansas City, MO 64108, USA; 4Department of Pediatrics, University of Kansas Medical Center, Kansas City, KS 66160, USA

**Keywords:** B cell receptor heavy-chain sequencing, vaccine response, SARS-CoV-2

## Abstract

Understanding the B cell response to SARS-CoV-2 vaccines is a high priority. High-throughput sequencing of the B cell receptor (BCR) repertoire allows for dynamic characterization of B cell response. Here, we sequenced the BCR repertoire of individuals vaccinated by the Pfizer SARS-CoV-2 mRNA vaccine. We compared BCR repertoires of individuals with previous COVID-19 infection (seropositive) to individuals without previous infection (seronegative). We discovered that vaccine-induced expanded IgG clonotypes had shorter heavy-chain complementarity determining region 3 (HCDR3), and for seropositive individuals, these expanded clonotypes had higher somatic hypermutation (SHM) than seronegative individuals. We uncovered shared clonotypes present in multiple individuals, including 28 clonotypes present across all individuals. These 28 shared clonotypes had higher SHM and shorter HCDR3 lengths compared to the rest of the BCR repertoire. Shared clonotypes were present across both serotypes, indicating convergent evolution due to SARS-CoV-2 vaccination independent of prior viral exposure.

## 1. Introduction

Severe acute respiratory syndrome coronavirus 2 (SARS-CoV-2) is the causal agent of Coronavirus disease 2019 (COVID-19) and continues to be a threat to human health across the globe [1,2]. SARS-CoV-2 mRNA vaccines are effective tools in the fight against this disease [3,4]. Although there is not a defined correlate of vaccine protection yet, high levels of SARS-CoV-2 spike neutralizing antibodies have been shown to correlate with vaccine efficacy [5,6]. While multiple cell types have critical roles in the adaptive immune response [7], here we focus specifically on the antibody-mediated humoral immune response to the SARS-CoV-2 vaccine generated in B lymphocytes. Subtle changes in B cell response can be captured by high-resolution B cell receptor (BCR) sequencing, including undescribed effects of prior COVID-19 infection on vaccination. 

B cells express transmembrane immunoglobulins, otherwise referred to as B cell receptors, which if secreted, recognize antigens as antibodies [8]. Collectively, all BCR sequences expressed by B cells constitute an individual’s BCR repertoire [9]. Diversity of the BCR repertoire is created by V(D)J recombination, a process by which genetic variability is made by altering the usage of gene segments: variable (V), joining (J), and diversity (D) regions [10]. To fine-tune the humoral immune response, activated B cells can enter germinal centers and undergo affinity maturation, resulting in further diversification. Affinity maturation involves two broad but related processes: somatic hypermutation (SHM) and clonal selection [11]. SHM induces mutations in the variable regions of immunoglobulins, and positive selection occurs for receptors with the highest antigen affinity, resulting in high-affinity antigen-specific B cell clones [12]. High-throughput sequencing of BCR repertoires, referred to as BCR-seq, can reveal many dynamic aspects of humoral immune responses including responses to both infection and vaccination [13]. 

Analyses of BCR repertoire responses to natural SARS-CoV-2 infection have revealed increased immunoglobulin isotypes IgA and IgM, and a targeted antibody response to the entire SARS-CoV-2 spike protein [14]. B cell response to SARS-CoV-2 infection has also been found to be marked by a slight increase in SHM (IgM/G/A) in the early stages, then decreasing and remaining low while a dominance of a few high-affinity B cell clones occurs [15]. Increased COVID-19 disease severity: use of ventilator and even death, has also been associated with the observation of elevated rates of SHM that do not decrease over time [16,17]. Meanwhile, in response to mRNA SARS-CoV-2 vaccination, increased isotype usage of IgG and a narrow SARS-CoV-2 receptor binding domain (RBD) antibody response have been observed [14]. In contrast to natural infection, SHM rates were not affected by SARS-CoV-2 vaccination [14], suggesting that differential BCR responses exist between infection and vaccination. Additionally, recent analyses have revealed common SARS-CoV-2 clonotypes present in different individuals, known as public clonotypes [18,19]. These public clonotypes signify that there is convergent evolution across BCR repertoires, and that B cell response to COVID-19, across multiple people may occur in a more similar way than previously thought. Here, we also investigate if convergent evolution occurred in our dataset and determine if public clonotypes are present. This will allow us to understand if SARS-CoV-2 vaccination elicits BCR public clonotypes. What is still unclear from these recent studies is how previous COVID-19 infection could impact BCR repertoire responses to SARS-CoV-2 vaccination. Characterizing the BCR response to vaccination by previous exposure will allow us to better understand how and if vaccines employ B cell memory and allow us to better understand how vaccines work.

## 2. Materials and Methods

### 2.1. Individuals and Sample Collection

We enrolled health care workers from our children’s hospital with no known history of SARS-CoV-2 infection (*n* = 4, seronegative) and with previous PCR-confirmed SARS-CoV-2 infection, 30–60 days prior to this study (*n* = 5, seropositive). Peripheral blood was collected prior to vaccination with Pfizer mRNA vaccine (Comirnaty^®^, Pfizer, New York, NY, USA) (week 0) and after primary immunization (week 3). All participants received only one dose of vaccine before BCR analyses. This cohort consisted of 9 individuals with an average age of 40.78 years, ranging from 28–58 years of age. All individuals were white, non-Hispanic or Latino, 7/9 were female, and 2/9 were male. Seropositive group had an average age of 42.4 years, and the seronegative group had an average age of 38.75 years. The seropositive group was all female, and the seronegative group was 2/4 male and 2/4 female. SARS-CoV-2 vaccine specimens were collected at Children’s Mercy Kansas City and were reviewed and approved by the Children’s Mercy IRB. Both plasma and peripheral blood mononuclear cells (PBMCs) were isolated in parallel from the blood collections and stored in ultra-low temperature freezers until use in antibody titer quantification and BCR repertoire sequencing, respectively.

### 2.2. Antibody Titers

To measure antibody levels to the SARS-CoV-2 spike protein subunits, spike subunit 1 (S1), spike subunit 2 (S2), receptor-binding domain (RBD), and nucleocapsid (NP), were used on a bead-based multiplex assay based on the Luminex (Austin, TX, USA) xMAP technology (HC19SERG1-85K-04, HC19SERA1-85K-04, HC19SERM1-85K-04, Millipore, Burlington, MA, USA). Reagent kits with secondary antibodies specific for isotypes IgG, IgM, IgA were used following the manufacturers protocol. The kit provided a set of SARS-CoV-2 antigen conjugated beads (S1, S2, RBD, NP) along with 3 positive control beads and a negative control bead set. The positive control beads were coated with different concentrations of IgG. The negative control beads did not have antigen conjugated to determine nonspecific binding. The 3 antigen-conjugated beads, 3 positive control beads, and 1 negative control beads were mixed and incubated with each plasma sample at a dilution of 1:100 with assay buffer. Samples were run in technical duplicate. To determine background activity, at least two sample wells per assay plate contained only buffer and no plasma. PE-anti-human IgG conjugate detection antibody was utilized to determine antibody response to each SARS-CoV-2 antigen. Using the positive control beads, inter-assay (plate-to-plate) co-efficient of variation (CV) was determined to be 5.16% for IgG. We utilized the Luminex analyzer (MAGPIX) and Luminex xPONENT acquisition software to acquire and analyze data. After acquisition net MFI was calculated by subtracting background MFI (no plasma).

### 2.3. RNA Extraction and Library Preparation

Frozen buffy coat PBMC samples derived from partitioned whole blood samples were processed using the RNeasy Plus Micro Kit (Cat# 74004, Qiagen, Germantown, MD, USA). Further, 350 ul of RLT buffer was added to buffy coat and lymphocytes were lysed via pipetting and homogenized with Qiashredder spin columns (Cat# 79656, Qiagen). gDNA eliminator columns were used to remove genomic DNA following homogenization. The remaining RNA extraction was carried out according to the RNeasy Plus Micro Kit protocol. RNA quality and quantity was determined using a nanodrop spectrophotometer (Cat# 13-400-519, ThermoFisher Scientific, Waltham, MA, USA). At least 25 ng of total RNA was inputted into the Archer BCR Library Prep for each sample. Archer Immunoverse-HS BCR Protocol for Illumina library prep was performed according to manufacturer instructions (ArcherDX, Palo Alto, CA, USA). Please see Appendix A for depth of sequencing pre- and post-quality control, RNA concentration at isolation, and number of unique clones per sample. Quality control procedures were followed as standard by Archer: minimum of 1.5 million reads per sample and 400–600 ng of RNA input, samples that did not follow this were removed from analysis. Libraries were quantified using the Kappa Library Quantification Kit (Illumina, San Diego, CA, USA). Libraries were pooled at an equimolar concentration of 4 nM. Libraries were sequenced on the Illumina MiSeq using 35% PhiX spike-in to account for low diversity, with the 2 × 300 base pair format. 

### 2.4. Data Analyses

Original raw FASTQ files were obtained using Archer Immunoverse-HS BCR Protocol. FASTQ files were processed through “Archer Immunoverse BCR IGH IGKL v1.0” pipeline for adaptor trimming and deduplication (Archer, San Jose, CA, USA). Bam file (sample.molbar.trimmed.deduped.merged.bam) obtained after completion of Archer Immunoverse pipeline was used to generate FASTQ file using samtools v1.10 (Genome Research Limited, Cambridge, UK). Fastq files for read1 (forward read) and read2 (reverse read) were created from a single fastq files obtained from merged bam file. Paired-End reAd mergeR (PEAR v0.9.6, Exelixis Lab, Heidelberg, Germany) was used to merge pair end reads. Merged pair end reads FASTQ file were converted to FASTA file.

MaskPrimers.py available on pRESTO—The Repertoire Sequencing Toolkit (v0.6.2, Kleinstein Lab, New Haven, CT, USA) was used for assigning the isotype information for each read as described in Immcantation portal section “Isotype and Primer Annotations”. IMGT database was used for V(D)J gene annotations and only the reads with functional heavy chain were kept for assigning clones and down streaming analysis. Reported ClonalAbundance, ClonalDiversity and Physicochemical was calculated using Alakazam (v1.1.0). Shazam (v1.0.2, Kleinstein Lab, New Haven, CT, USA) was utilized for calculating Mutational count. Default settings were used for samtools and PEAR whereas similar settings were utilized for running pRESTO, Change-O, Alakazam and Shazam as described in Immcantation Portal.

To determine the expanded clone set we thresholded the data by using the 50 most numerous IgG clones by read number for each individual at week 3. See Appendix A for top 50 clones at each time point and overlap between time points. Complementarity-determining region 3 (HCDR3) sequences were queried in the COVID-antibody-database (Cov-ab-dab) [20]. 

### 2.5. Statistical Tests

All statistical tests (*t*-test, Mann–Whitney, Wilcoxon, Kolmogorov–Smirnov) were conducted in Prism 9 (Graph Pad, San Diego, CA, USA). 

### 2.6. Data Availability

All sequencing data will be made publicly available through NCBI, accession PRJNA839082.

## 3. Results

### 3.1. BCR-seq of Peripheral Blood after COVID-19 Vaccine 

We previously determined that antibody titers to SARS-CoV-2 spike protein in response to SARS-CoV-2 vaccination are higher in individuals with a history of recent COVID-19 infection (seropositive) when compared to those without (seronegative) [21,22,23,24]. From this dataset, we selected five seropositive and four seronegative individuals for BCR-seq from weeks 0 and 3 (Figure 1A). Antibody levels of isotypes IgG, IgM, and IgA in blood plasma from these individuals to SARS-CoV-2 proteins: spike protein 1 (S1), spike protein 2 (S2), receptor binding domain (RBD) and nucleocapsid (NP) were detected with IgG levels being the highest detected and IgA levels being the lowest (Figure 1B). Generally, antibody titers were higher for seropositive compared to seronegative at week 0. At week 3, seropositive IgG levels remained higher than seronegative for S1, S2, and NP, but RBD levels were similar (Figure 1B). At week 3, IgM levels for S2 were higher in seropositive group, while S1, RBD, NP had no significant differences between groups (Figure 1B). At week 3, differences in IgA antibody titers between seropositive and seronegative were not significant for any of the antigens. Overall, these data indicated that within our selected dataset, similar to previous reports, that COVID-19 infection conferred higher antibody levels in response to the first dose of vaccine. 

### 3.2. Immunization Did Not Alter the Global BCR Isotype, Variable Gene Usage, or HCDR3 Length Distribution

At baseline before immunization (week 0), IgM sequences had the highest frequency of isotype for both seropositive and seronegative individuals (seropositive: 66.78%, seronegative: 61.45%), followed by IgG (21.55%, 21.16%), IgD (8.7%, 12.24%), IgA (2.91%, 5.08%), and IgE (0.06%, 0.07%) (Figure 2A). After a single SARS-CoV-2 immunization (week 3), no significant changes in isotype usage were observed in the BCR repertoire (Figure 2A). There were also no differences in isotype usage between seropositive and seronegative groups at either timepoint.

We then focused on the BCR sequences that contained the IgG isotype as these represented the subclass that had the highest levels in the peripheral blood to SARS-CoV-2 (Figure 1B). First, we analyzed the frequency of the variable gene family (IGHV) usage. IgG BCR sequences utilized variable gene family 3 (IGHV3) at the highest frequency, followed by V4, V1, V5, V2, V6, and V7 (Figure 2B). No significant differences were observed in the frequencies of V gene usage between seropositive and seronegative groups at week 0 or week 3 (Figure 2B). When we analyzed V gene usage for IgM (Appendix A) or IgA in BCR sequences (Appendix A) we found similar results, with no significant differences found in the frequencies of V gene usage between seropositive and seronegative groups at weeks 0 or 3. These data suggested the total BCR isotype usage and IgG/M/A V gene usage were similar before and after vaccination and were not different in individuals infected with COVID-19 before vaccination. 

No significant changes in the distribution of heavy-chain complementarity determining region 3 (HCDR3) lengths between seropositive and seronegative at week 0 (Kolmogorov–Smirnov Test; *p* = 0.9525), or at week 3 (Kolmogorov–Smirnov Test; *p* = 0.9983) were observed (Figure 2C). Similarly, the distributions of HCDR3 lengths in IgM and IgA were not different between seropositive and seronegative at week 0 or week 3 (Kolmogorov–Smirnov Test; week 0 *p* = 0.9525; week 3 *p*= 0.9983 and week 0 *p* = 0.9983; week 3 *p*= 0.9983, respectively) (Appendix A). Therefore, our data suggested that HCDR3 lengths within the whole BCR repertoire were not globally altered by previous infection or vaccination (first dose).

### 3.3. BCR SHM Increased after Vaccination in the Seropositive Group and Decreased in Seronegative Group

We compared levels of IgG SHM between groups prior to (week 0) and 21 days after first dose of vaccine (week 3). We found that a similar proportion of clones that had ≥2 mutations or had <2 mutations at both time points, with no significant difference in proportions between groups (Figure 3A, Appendix A). Within IgG BCR sequences with ≥2 mutations, we found that the seronegative group had higher SHM at baseline and week 3 (*t*-test with Welch’s correction; *p* < 0.0001; *t*-test with Welch’s correction; *p* = 0.0002) (Figure 3B). When comparing SHM across time points, we observed a significant increase in BCR SHM at week 3 compared to week 0 in the seropositive group (*t*-test with Welch’s correction; *p* < 0.0001) (Figure 3B). Conversely, there was a significant decrease in BCR SHM at week 3 when comparing week 0 to in the seronegative group (*t*-test with Welch’s correction; *p* < 0.0001) (Figure 3B).

IgM SHM followed a similar pattern as IgG, whereby the seropositive group increased SHM at week 3, while seronegative group decreased (*t*-test with Welch’s correction; seropositive *p* < 0.0001; seronegative *p* < 0.0001) (Appendix A). SHM in IgA increased for seropositive from week 0 to week 3 (*t*-test with Welch’s correction; *p* < 0.0001) while for the seronegative group, no significant difference was observed between week 0 and week 3 (*t*-test with Welch’s correction; seronegative *p* = 0.0735) (Appendix A). Altogether, these results indicated modest differences in the frequencies of SHM in the IgG, IgM and IgA where SARS-CoV-2 vaccination induced increases in SHM in seropositive group and decreases or results in no change to SHM in seronegative group. 

To determine if BCR IgG repertoires were more diverse in response to vaccination we conducted diversity analyses of the B cell clones using 3 measures: species richness, Shannon diversity index, and Simpson’s diversity index. No statistically significant differences in species richness, Shannon diversity, Simpson’s diversity were observed for the B cell clones identified between seropositive and seronegative groups at week 0 (Figure 3C; Mann–Whitney test; *p* > 0.9999, Mann–Whitney test; *p* > 0.9999, Mann–Whitney test; *p* = 0.6857). At week 3, there was a trend for BCR clones from the seropositive group to have greater species richness and Shannon diversity than the seronegative group (Mann–Whitney test; *p* = 0.0635, Mann–Whitney test; *p* = 0.1905). Overall, BCR diversity trended toward being higher in the seropositive group compared to seronegative group at week 3, after first vaccine dose. 

### 3.4. Altered Genetic Features of the Most Abundant Clonotypes between Seropositive and Seronegative Groups

When we examined pre-existing IgG clones in the repertoire at week 3, we found that the majority of the IgG clone repertoire was made of novel clones (Figure 4D, top panel). At week 3, for the seropositive group on average 98.2% of the repertoire was made of novel clones, and for the seronegative group on average 98% clones were novel. Pre-existing IgG clones in the repertoire at week 3 are minimal (ranging 5.8–0.67%, average seropositive 1.8%, average seronegative 2%). We then selected the 50 most abundant IgG clones as the most numerous clonotypes based on reads from each sample at week 3. This threshold was determined by reviewing clonal frequency by reads at week 3 (Appendix A) and referring to previous literature where the most numerous clones within the repertoire were thresholded and analyzed [16]. There was minimal overlap between top 50 most numerous IgG clonotypes at week 3 and clones of any isotype at week 0 (Figure 4D, bottom panel). For seropositive samples, on average 89.6% of the top 50 clones were novel at week 3. For seronegative samples, on average 74.5% were novel at week 3. This indicated that most of the top 50 IgG clones at week 3 were novel at week 0. Furthermore, we analyzed the top 50 most abundant IgG clones at week 0 and found minimal overlap with the top 50 IgG clones at week 3 (Appendix A). For seropositive, on average 93% of the top 50 at week 0 were not present in week 3 top 50 and for seronegative 95.2% were not present in week 3 top 50. This indicated that very few clones were in the expanded group at both time points. 

V gene usage in the top 50 clone groups did not significantly differ from the remaining, less abundant clones “other” (Figure 4A). Furthermore, we did not observe any significant difference in the IGHV gene usage in the Top 50 clones between seropositive and seronegative groups. 

The average HCDR3 length was statistically significantly shorter in the top 50 group of BCR clones when compared to all other clones for both seropositive (Mann–Whitney test, *p* = 0.0079) and seronegative (Mann–Whitney test, *p* = 0.0286) groups (Figure 4B). There were no statistical differences when comparing top 50 HCDR3 length mean groups between serotypes (Mann–Whitney test; *p* = 0.2857) (Figure 4B). The distribution of HCDR3 lengths for top 50 was also significantly different when compared to the other clone group (Appendix A; seropositive, Kolmogorov–Smirnov *p* = 0.0354; seronegative, Kolmogorov–Smirnov *p* = 0.0354). Top 50 expanded clones in response to vaccination in both serotypes had shorter HCDR3 lengths compared to other clones. 

SHM was higher in the seropositive top 50 clones (mean 4.044%) when compared to the top 50 seronegative clones (mean 3.541%; *t*-test with Welch’s correction; *p* = 0.0080) (Figure 4C). Seronegative top 50 clones had lower SHM when compared to other clones (*t*-test with Welch’s correction; *p* = 0.0123) (Figure 4C). No statistically significant difference in SHM was observed between top 50 clones and other clones in the seropositive group (Figure 4C). Overall, these data indicated that seropositive expanded clones had higher SHM than seronegative expanded clones in response to vaccination. 

To investigate whether IgG clones from previous infection were expanded in response to the vaccine, we determined, within the expanded top 50 clone group, if any of the clonotypes were present at week 0 before vaccination. When comparing IgG clones from the whole BCR repertoire at week 3 and to week 0, pre-existing clones only accounted for 1.7% in the seropositive and 3.4% in the seronegative, which was not statistically significant (Figure 4D, top panel). Usage of pre-existing IgG clones in the top 50 group averaged 12.8% in the seropositive group and 25% in the seronegative group (Figure 4D, bottom panel). We characterized the top 50 clones at time point 1 and compared to the top 50 at time point 2 and found little overlap (Appendix A). We did not see a statistically significant difference between seropositive and seronegative groups in usage of pre-existing clones. Both seropositive and seronegative had pre-existing clonotypes that were expanded in number after immunization, but both repertoires were dominated by novel clones at week 3 (Figure 4D).

### 3.5. Convergent Clones Were Observed across Both Serotypes including 28 Clones Present in All Samples in This Study

We determined shared clone usage based on highly similar HCDR3 sequences present between two or more individuals and limited our scope to the 50 most numerous clones. We identified 28 clonotypes that were present across all individuals (*n* = 9, S1–9) in this study (Figure 5A). Seven clones were present in two individuals (*n* = 2). Two clones were shared between two individuals in three instances (*n* = 2). One clone was shared across two individuals in three instances (*n* = 2), one clone was shared across three individuals in three instances (*n* = 3), one clone was shared across four individuals in one instance (*n* = 4), and one clone was shared across six individuals in two instances (*n* = 6) (Figure 5A). No detectable difference in convergent clones was determined between the serotypes. 

We then characterized the properties of the 28 convergent clones that were shared across all 9 individuals: HCDR3 length, V gene usage, and SHM. No difference was observed between the 28 convergent seropositive (mean = 7.617) and seronegative groups (mean = 7.500) (*t*-test with Welch’s correction *p* = 0.8975). SHM was higher in the 28 convergent clone groups when compared to the top 50 expanded clone group (*t*-test with Welch’s correction; *p* < 0.0001) and the remaining other clones in the repertoire (*t*-test with Welch’s correction; *p* < 0.0001) (Figure 5B). The 28 convergent clone group had the highest SHM of any clone group observed in this study (convergent 28: 7.4027%; top 50: 3.813%; other clones: 1.515%). HCDR3 length distribution was different from the remaining other clones in the repertoire, overall convergent clones had shorter HCDR3 lengths (Kolmogorov–Smirnov test; *p* = 0.0004) (Figure 5C). The HCDR3 length distribution was not statistically different between the 28 convergent and the top 50 expanded groups (Figure 5C). Overall, the 28 shared clone group had shorter HCDR3 lengths and higher SHM compared to other groups and convergent clones were present across both serotypes. 

### 3.6. Queried Clonotypes Included Matches to the COVID Antibody Database

To determine whether the convergent and expanded clones identified in this study were previously identified as COVID-specific clones, we queried the HCDR3 regions of the top 50 expanded clones from each individual and all convergent clonotypes to the COVID antibody database [20] (Table 1). We defined a match as greater than or equal to 80% alignment. We uncovered three matches to the database, including one of the twenty-eight convergent clones. Clone #34727 matched to antibody S-B8 in the database. This clone had a HCDR3 length of 10, used V3-11 and SHM was 7.626% (Table 1). The aligned antibody, S-B8, has reported SARS-CoV-2 neutralizing activity [25]. Clone #13327 aligned 80% to Fab-368 but was only present in one individual studied. Clone #13327 used V3-74, had a HCDR3 length of 10, and SHM was 1.670% (Table 1). The antibody Fab-368 also has been reported as neutralizing to SARS-CoV-2 [26]. Clone #8269 matched to a wide variety of targets including 100% alignment to: XG001, R121-3G10, R410-3D10, PDI-38, COVA1-27, H712443+K711941, R259-1F4, CV10, H712427+K71111927, and Shiakolas_53181-5. This clone used V4-59, had a HCDR3 length of 6, and a SHM of 0.759% (Table 1). These aligned antibodies were not reported to have neutralizing activities [27,28,29,30,31,32]. Clone #8269 also had 83.33% alignment with XG002, R616-1E6, PDI-124, R849-3B4, R849-1F9, PDI-211 (Table 1). These antibodies also did not have known neutralizing activities [27,28,29]. Clone #8269 has some cross-reactivity with SARS-CoV-1, binds the spike protein, and has been isolated from multiple human SARS-CoV-2 patients and vaccinees [28,33,34]. The short HCDR3 length of 6 amino acids could play a role in the alignment to multiple BCR sequences. These results identified three public clonotypes that were previously reported as SARS-CoV-2-specific in other studies, with two that had SARS-CoV-2 neutralization as a previously described function. 

## 4. Discussion

To understand how previous SARS-CoV-2 infection impacts the B cell receptor repertoire in response to vaccination, we sequenced and analyzed BCR repertoires of seropositive and seronegative individuals after the first dose of Pfizer COVID-19 mRNA vaccine. Although both T and B cells have crucial roles in adaptive immunity, our aim was to characterize cellular responses in B cells to explain differential antibody responses that have been observed after the first dose of vaccine. Our group and others have demonstrated that individuals with prior COVID-19 had higher levels of antibodies after the primary SARS-CoV-2 vaccination [21,23,24,35,36,37,38].

Overall, the BCR repertoire after a single SARS-CoV-2 vaccination at week 3 did not differ greatly based on prior SARS-CoV-2 infection before immunization. At week 0, the seronegative group had higher SHM compared to the seropositive group, this could be due to a decreased SHM rate in seropositive group, which has been observed in recently COVID-19 recovered individuals [15]. To a small degree, SHM increased in the seropositive group and decreased or remained the same in the seronegative group. It has been previously shown that SHM does not change in response to SARS-CoV-2 vaccination [14]. Here, we observed a decrease (IgG and IgM) or no change (IgA) for the seronegative group, which is in line with these previous findings. The increase in SHM observed in the seropositive group could indicate a reactivation and expansion of memory B cells which themselves have high SHM [15]. Increased SHM in the BCR in response to vaccination after a previous infection has previously been shown for influenza [39]. Additionally, when looking at SARS-CoV-2 recovered individuals after immunization, IgG+ memory B cells significantly increased after the first vaccine dose [40]. 

Although changes in the frequency of isotype usage have previously been observed in response to SARS-CoV-2 infection and vaccination, no changes in proportion of isotype usage were found in this study [14,17,27,41]. Increases in the proportion of IgG have been observed 25 days post-vaccination [14]. It is possible that a low sample size obscured seeing a difference in this study, and future studies of larger numbers of individuals should examine differential isotype usage between seropositive and seronegative individuals undergoing SARS-CoV-2 vaccination. 

We found that the majority of the BCR repertoire at week 3 was made up of novel clones. Pre-existing clones were only a minority of the most abundant (top 50) IgG clonotypes detected after vaccination in both serogroups. Furthermore, a minimal number of IgG clones were shared between top 50 at weeks 0 and 3. This indicated that expansion and selection of SARS-CoV-2 clonotypes in the seropositive group was not necessarily limited to pre-existing immunological memory from infection. In fact, expansion of novel clonotypes in response to vaccination is seen in both serotypes. 

We defined and characterized an expanded clone set to include only the most numerous clones within each repertoire. We observed that in the expanded clone set the seropositive group had significantly higher SHM than the seronegative group. This could be explained by an expansion of different B cell types. Specifically, higher SHM in the seropositive group could reflect an expansion of memory B cells. Expansion of plasmablasts (unmutated B cells with low levels of SHM) containing neutralizing antibodies with close to germ-line expression patterns (i.e., no mutations) have been observed early in naïve response to SARS-CoV-2 infection [31]. This could account for the SHM response pattern observed in the seronegative group. Shorter HCDR3 lengths have been associated with antigen exposure [42], and memory B cells have shorter HCDR3s [43]. It is possible that expanded clone groups are made up of more HCDR3s from memory B cells, making the HCDR3 lengths overall shorter. However, shorter HCDR3 lengths were observed in both expanded clone groups. It remains to be determined whether the differences in the features of the expanded clone groups are based on B cell types.

Highly convergent clones were found in our dataset in response to vaccination. These clones were present across both serotypes, and 28 of these clones appeared in every individual in this study. This suggests similar responses to vaccination across serotypes. Early clonal convergence in response to SARS-CoV-2 is a feature that has been previously described in other datasets [18,44]. These 28 convergent clones were unique as a group and had high levels of SHM and shorter HCDR3 lengths when compared to expanded and “other” clone groups, respectively. Use of the publicly available COVID antibody database revealed that three clones from our entire expanded clone set aligned to targets with greater than or equal to 80% homology of characterized SARS-CoV-2- specific B cells. Two of these public clonotypes (clone #34727, #13327) were previously found to generate SARS-CoV-2 neutralizing antibodies. The other public clone (clone #8269) had cross-reactivity with SARS-CoV-1 spike protein but was non-neutralizing. Functional studies of clone #8269 would be of future interest. These findings indicated that public clonotypes were generated in response to SARS-CoV-2 vaccination. 

Overall, our findings show that many features of the BCR repertoire in response to SARS-CoV-2 vaccination were largely similar in individuals with prior COVID-19 before immunization compared to seronegative individuals (V gene usage, HCDR3 length, diversity). However, previous infection did impact rates of SHM in response to vaccination. Specifically, higher SHM was seen in the seropositive expanded clone group after vaccination, perhaps indicating differential B cell maturation states between groups. Despite higher levels of SHM, individuals with prior infection had a majority of novel clonotypes expanded after immunization and did not utilize a majority of pre-existing clonotypes detected at baseline. Lastly, the presence of public clonotypes in our dataset and the public COVID database, indicated convergent B cell clonal evolution that could be harnessed across multiple individuals by SARS-CoV-2 vaccination.

The limitations of our study are that a small sample size was used, and that the majority of samples were from white, middle-aged females. Further study into the BCR repertoires post-primary vaccination, of other ethnicities, ages, and creating an expanded sample size would be of interest. Another limitation of this study is that expanded clonotypes are not antigen specific, we determined “expanded or “vaccine-induced” based on the number of reads at week 3. While the expansion of the clonotypes is likely to reflect their importance during vaccination, they may not be specific to SARS-CoV-2 vaccination response. Further confirmation of antigen specificity of these vaccine-induced/expanded clonotypes could be done to confirm their status. 

## Figures and Tables

**Figure 1 vaccines-10-01477-f001:**
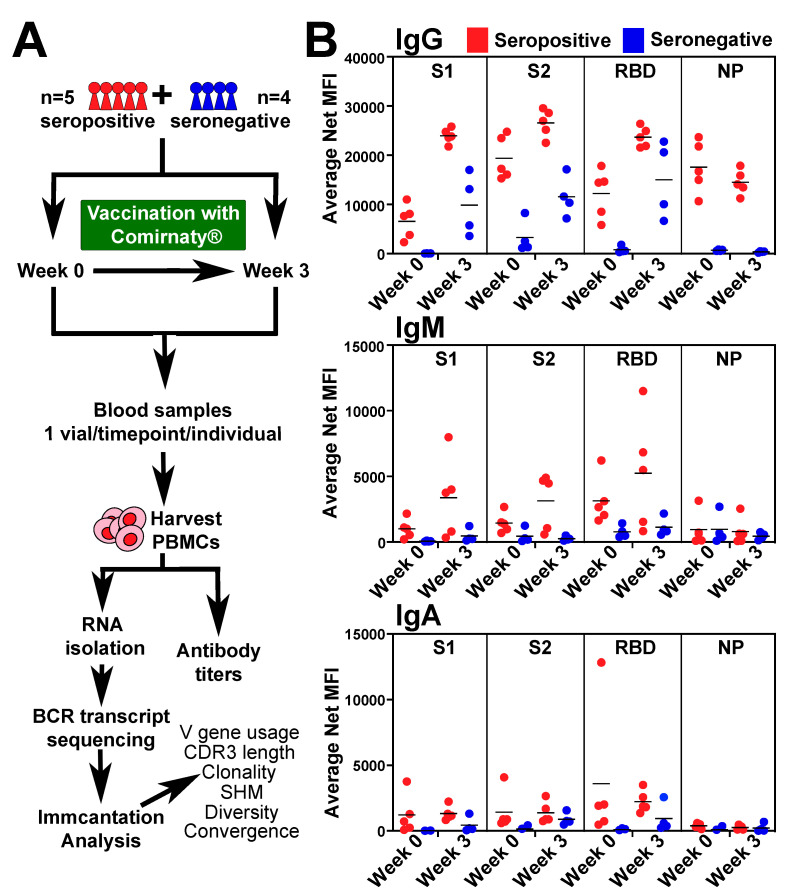
**Experimental design and antibody titers in response to vaccination.** (**A**) Schematic showing experimental design and details. (**B**) Antibody titers IgG/M/A for SARS-CoV-2 spike 1 protein (S1), spike 2 protein (S2), receptor binding domain (RBD), and nucleocapsid protein (NP) of seropositive (red, *n* = 5) and seronegative (blue, *n* = 4) individuals: pre-vaccination (week 0) and 21 days after (week 3) first dose of Comirnaty^®^ (Pfizer, New York, NY, USA).

**Figure 2 vaccines-10-01477-f002:**
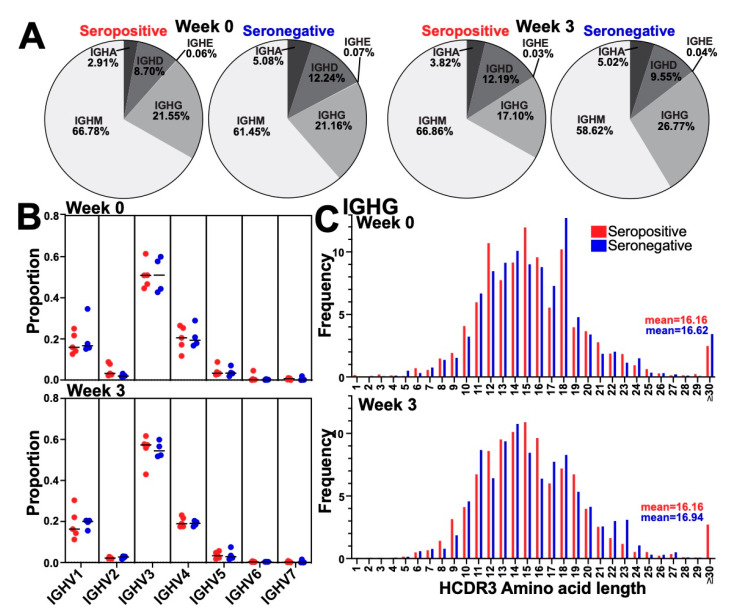
**Isotype usage, V gene usage, and HCDR3 length of BCR repertoire across serotypes during vaccination.** (**A**) BCR isotype usage (IgM IgG, IGHD, IgA, and IGHE) for seropositive and seronegative individuals at each time point (week 0, week 3): percentages as indicated. No changes were seen in the proportion of any isotype due to vaccination, nor significant variation observed between serotypes. (**B**) IgG Heavy chain V gene usage (IGHV1-7) of IgG clonotypes. Proportion as indicated for seropositive (red) and seronegative (blue). No significant differences were observed between serotypes at week 0 and week 3, and no changes within serotype over time were observed. (**C**) Distribution of IgG heavy chain complementary determining region 3 (HCDR3) lengths of IgG clones for week 0 and week 3. Seropositive (red) and seronegative (blue) and mean HCDR3 length are shown. No significant changes in the distribution of HCDR3 lengths were seen between serotypes at either time point, or within serotype between weeks.

**Figure 3 vaccines-10-01477-f003:**
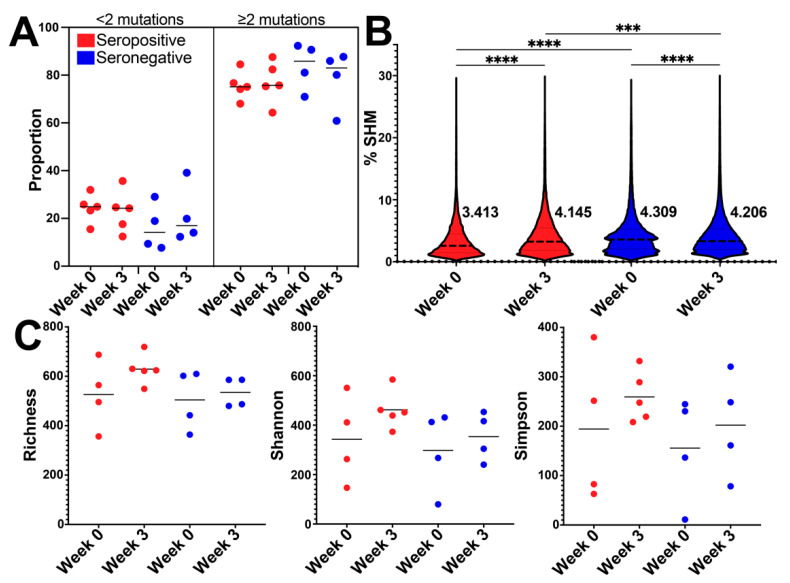
**Somatic hypermutation and diversity measures of B cell receptor (BCR) repertoire across serotypes during vaccination****.** (**A**) Plot depicting the proportion (0–100%) of IgG clones with less than 2 mutations (left) and greater than 2 mutations (right). There were no statistically significant differences in the proportions of seropositive (red) and seronegative (blue) at week 0 or week 3. (**B**) Violin plots depicting percent of SHM for IgG clonotypes in seropositive (red) and seronegative (blue) at weeks 0 and 3. Mean values for each violin are indicated on the plot and by strong dashed lines, quartiles are indicated by thin dashed lines. Statistical test performed was Welch’s *t*-test, *p* value **** *p* < 0.0001, *** *p* = 0.0002. (**C**) Diversity measures plots including species richness, Shannon diversity, and Simpson diversity indexes. Seropositive is indicated by red, and seronegative by blue and time points: week 0 and week 3.

**Figure 4 vaccines-10-01477-f004:**
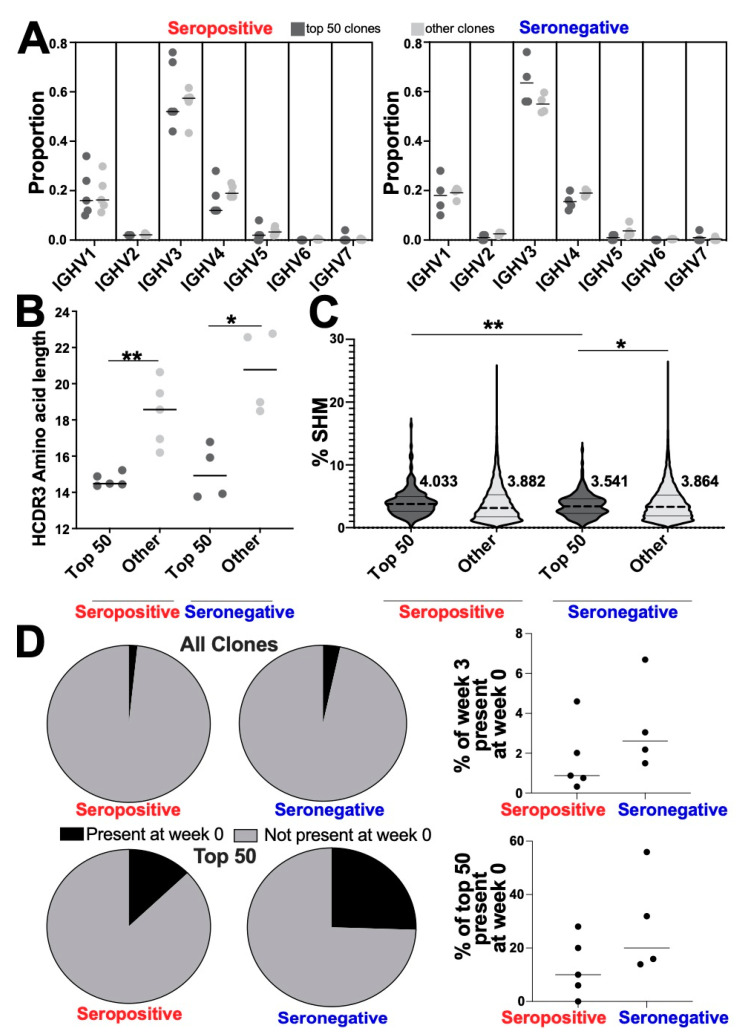
**Characteristics of expanded “top 50” IgG clonotypes in seropositive and seronegative individuals after one dose of Comirnaty^®^ (****Pfizer, New York, NY, USA).** (**A**) IGHV gene usage plotted for top 50 IgG clonotypes compared to the remaining clones in the repertoires of seropositive (left) and seronegative (right) individuals at week 3. No differences were observed in the V gene usage patterns between the clone groups (top 50 v. other) in either serotype. (**B**) Top 50 IgG clones’ HCDR3 length was shorter than “other” (the remaining clonotypes) for both seropositive and seronegative (Mann–Whitney test ** *p*= 0.0079, * *p* = 0.0286). No difference in top 50 HCDR3 length was observed between serotypes. (**C**) Violin plots depicting percentage SHM for top 50 IgG clones and other based on serotype, strong dashed line indicates mean value, fine dashed line indicates quartile, means are to the right of each violin. (Welch’s *t*-test; ** *p* = 0.0080, * *p* = 0.0123). (**D**) Pie charts depicting the proportion of IgG clones that were present at week 0 (black) and novel at week 3 (gray) for the entire repertoire (top) or just top 50 (bottom). Serotype indicated under pie charts. Right panel, plots depicting individual proportions of clones present at week 0 for the whole repertoire (top) and top 50 (bottom). No statistical differences were observed between serotypes.

**Figure 5 vaccines-10-01477-f005:**
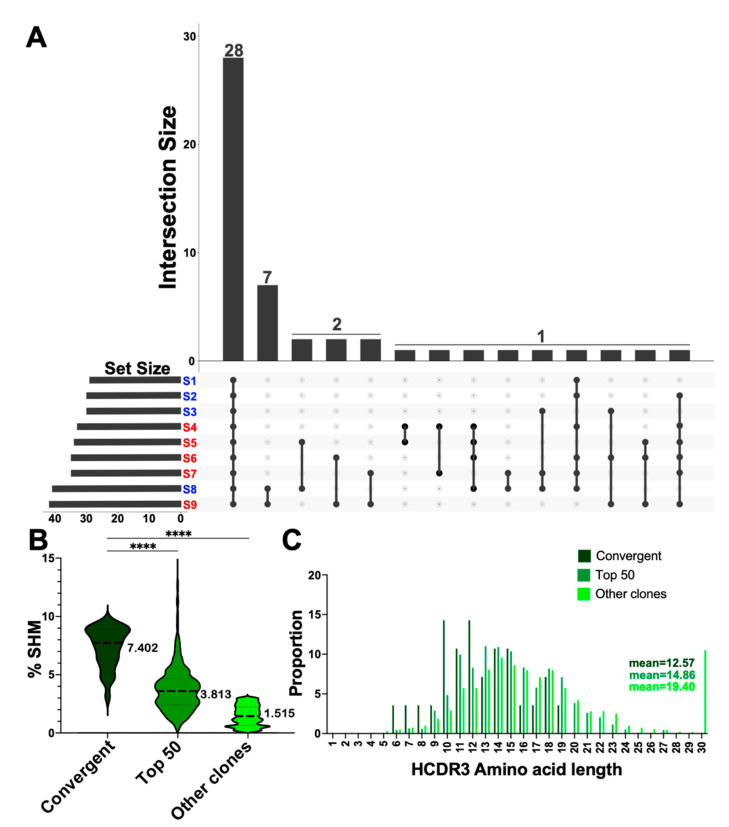
**Convergent clones present at week 3.** (**A**) Upset plot depicting convergent clones. Intersection size (y axis) indicates number of clones, each bar is demarcated with number of unique clones. Connected dots below each bar indicate the number of individuals in which those clone(s) are present. Set size includes the total convergent clones for each sample. S1–9 indicate samples 1–9 red (seropositive) and blue (seronegative) (**B**) SHM percent comparison between the 28 convergent clones seen in (**A**), top 50 expanded clones and all other clones. Welch’s *t*-test, **** *p* < 0.0001. (**C**) HCDR3 length distribution of the 28 convergent clones (dark green), top 50 collapsed by serotype (jungle green), and other clones (lime green), means are indicated on the plot by color. Kolmogorov–Smirnov test, convergent vs. other, *p* = 0.0004.

**Table 1 vaccines-10-01477-t001:** Features of convergent clones that aligned to public clonotypes.

Clone	Alignment Target	Sequence Match %	Number of Individuals Detected	VDJ Gene	CDR3 Length	Mean SHM %	Neutralizing
#34727	S-B8	80	9	IGHV3-11*06	10	7.626758896	Yes
#13327	Fab-368	80	1	IGHV3-74*01	10	1.670646982	Yes
#8269	XG001	100	1	IGHV4-59*09	6	0.758942973	No
#8269	R121-3G10	100	1	IGHV4-59*09	6	0.758942973	No
#8269	R410-3D10	100	1	IGHV4-59*09	6	0.758942973	No
#8269	PDI-38	100	1	IGHV4-59*09	6	0.758942973	No
#8269	COVA1-27	100	1	IGHV4-59*09	6	0.758942973	No
#8269	H712443+K711941	100	1	IGHV4-59*09	6	0.758942973	No
#8269	R259-1F4	100	1	IGHV4-59*09	6	0.758942973	No
#8269	CV10	100	1	IGHV4-59*09	6	0.758942973	No
#8269	H712427+K711927	100	1	IGHV4-59*09	6	0.758942973	No
#8269	Shiakolas_53181-5	100	1	IGHV4-59*09	6	0.758942973	No
#8269	XG002	83.33	1	IGHV4-59*09	6	0.758942973	No
#8269	R616-1E6	83.33	1	IGHV4-59*09	6	0.758942973	No
#8269	PDI-124	83.33	1	IGHV4-59*09	6	0.758942973	No
#8269	R849-3B4	83.33	1	IGHV4-59*09	6	0.758942973	No
#8269	R849-1F9	83.33	1	IGHV4-59*09	6	0.758942973	No
#8269	PDI-211	83.33	1	IGHV4-59*09	6	0.758942973	No

## Data Availability

Sequence data will be accessible from 30 August 2022. https://www.ncbi.nlm.nih.gov/sra/PRJNA839082.

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
