# Peer review of "Effects of Prior Infection with SARS-CoV-2 on B Cell Receptor Repertoire Response during Vaccination"

_vaccines, 2022, doi:10.3390/vaccines10091477_

Round 1

Reviewer 1 Report

The manuscript entitled "Effects of Prior Infection with SARS-CoV-2 on B Cell Receptor 2 (BCR) Repertoire Response during Vaccination " provides data on diversity of BCR sequences in seropositive and seronegative individuals after the vaccination with Pfizer SARS-CoV-2 mRNA vaccine. The study lacks information on the deepness of the sequencing and also parallel RNA isolation from each individual to assess the saturation of the sequencing data. Without this the data has very limited information value.

The manuscript also lacks information on the number of clones identified in each individual. The number of all clones should be also shown in the Figure 4D.

Authors do not provide any proof that the pre-existing clones found in the "Top 50" group were not dominant also before the vaccination. Many or most of them may not have any connection with SARS-CoV-2 vaccination/infection.

Reviewer 2 Report

In this paper, the authors characterized and compared BCR usage in seropositive and seronegative healthy workers using the bulk BCR sequencing. They concluded that the majority of BCR are similar between these two groups. The conclusion is mostly due to the fact there are pre-existing IgG/IgM/IgA B cells induced by  previous infections or immunizations, which is the major limitation of this study.  Statistical analysis without knowing the antigen specificity is mis-leading.

    I would suggest the authors compare the B cells clones from the same people before and after COVID-19 mRNA vaccination to investigate the B cell response to the vaccination. By subtracting the pre-existing IgG/IgM/IgA B cells, they should be able to specifically look at those B cells responding to the  COVID-19 mRNA vaccination.

Author Response

We thank the reviewers for their time and consideration in reviewing our manuscript.  With this revision, we have addressed each of their comments and suggestions. Below is a point-by-point response to each reviewer comment. We believe this has greatly improved our study resulting in a stronger manuscript with increased clarity and scientific precision.

Reviewers' Comments:

Reviewer #2:

  1. In this paper, the authors characterized and compared BCR usage in seropositive and seronegative healthy workers using the bulk BCR sequencing. They concluded that the majority of BCR are similar between these two groups. The conclusion is mostly due to the fact there are pre-existing IgG/IgM/IgA B cells induced by previous infections or immunizations, which is the major limitation of this study. 

RESPONSE: We thank the reviewer for this comment and have addressed the same concern in response to reviewer 1. We found that the majority of the IgG clone repertoire was made of novel clones (Figure 4D, top panel). Pre-existing IgG clones were minimal in the repertoire. Additionally, the majority of “top 50” expanded IgG clones at week 3 were not present at week 0 in any isotype (Figure 4D, bottom panel).  Furthermore, we re-analyzed our data to look at the top 50 IgG clones at time point 1 to see if they were present at week 3 or in the top 50 group at week 3. We then created Supplemental Table S2 and added that to the manuscript and described those findings. The top 50 groups of weeks 0 and 3 have very little overlap.

The results section has been updated to better explain these findings and clarify the findings.

“When we examined pre-existing IgG clones in the repertoire at week 3, we found that the majority of the IgG clone repertoire was made of novel clones (Figure 4D, top panel). At week 3, for the seropositive group on average 98.2% of the repertoire was made of novel clones, and for the seronegative group on average 98% clones were novel. Pre-existing IgG clones in the repertoire at week 3 are minimal (ranging 5.8 - 0.67%, average seropositive 1.8%, average seronegative 2%). We then selected the 50 most abundant IgG clones as the most numerous clonotypes based on reads from each sample at week 3. This threshold was determined by reviewing clonal frequency by reads at week 3 (Supplemental Figure 3) and referring to previous literature where the most numerous clones within the repertoire were thresholded and analyzed. There was minimal overlap between top 50 most numerous IgG clonotypes at week 3 and clones of any isotype at week 0 (Figure 4D, bottom panel). For seropositive samples, on average 89.6% of the top 50 clones were novel at week 3. For seronegative samples, on average 74.5% were novel at week 3.  This indicated that most of the top 50 IgG clones at week 3 were novel at week 0. Furthermore, we analyzed the top 50 most abundant IgG clones at week 0 and found minimal overlap with the top 50 IgG clones at week 3 (Supplemental Table S2). For seropositive, on average 93% of the top 50 at week 0 were not present in week 3 top 50 and for seronegative 95.2% were not present in week 3 top 50. This indicated that very few clones were in the expanded group at both time points.”

  1. Statistical analysis without knowing the antigen specificity is mis-leading.

RESPONSE: The reviewer brings up an excellent point. We determined expanded clones based on read number and not antigen-specificity. This may be an excellent avenue to follow up on in future studies, particularly of the clones we identified as expanded and convergent. The limitations have been addressed in the discussion.

“The limitations of our study are that a small sample size was used, and that the majority of samples were from white, middle-aged females. Further study into the BCR repertoires post-primary vaccination, of other ethnicities, ages, and creating an expanded sample size would be of interest. Another limitation of this study is that expanded clonotypes are not antigen specific, we determined “expanded or “vaccine-induced” based on the number of reads at week 3. While the expansion of the clonotypes is likely to reflect their importance during vaccination, they may not be specific to SARS-CoV-2 vaccination response. Further confirmation of antigen specificity of these vaccine-induced/expanded clonotypes could be done to confirm their status.”

  1. I would suggest the authors compare the B cells clones from the same people before and after COVID-19 mRNA vaccination to investigate the B cell response to the vaccination. By subtracting the pre-existing IgG/IgM/IgA B cells, they should be able to specifically look at those B cells responding to the COVID-19 mRNA vaccination.

RESPONSE: Subtracting pre-existing clones from each sample is a good suggestion. However, when we looked at the occurrence of pre-existing IgG clones at week 3, we found that they were minimal (ranging 5.8 - 0.67%, average seropositive 1.8%, average seronegative 2%) (Figure 4D, top panel).  A similar analysis for pre-existing clones for IgM and IgA could be done, but for this manuscript we focused on the IgG isotype due to its high concentration in circulating blood (Figure 1B). Additionally, the week 3 top 50 IgG clone groups were made of a majority of novel clones (89.6%- 74.5%), and there was little overlap between top 50 groups at week 0 and 3 (7-4.8%) (Figure 4D, bottom panel).

Reviewer 3 Report

In this manuscript titled “Effects of prior infection with SARS-CoV-2 on B cell receptor (BCR) repertoire response during vaccination”, the authors collected PBMC samples from SARS-CoV-2 seropositive or seronegative donors before and after vaccination, and analyzed the characteristics of their BCR repertoire, including BCR isotypes, V gene usage, HCDR3 length, SHM and the distribution of common clones. The authors found that there was a minor difference between seropositive and seronegative donors after vaccination and there were 28 common clones could be found across all the individual donors. Overall, this is an interesting and timely study. The manuscript is very well written. I only have several minor comments.

Minor issues:

1. It seems that all of the donors only got one dose of vaccination by week 3. If this is the case, please explicitly state it in the method section.

2. Line 235-237, “Within IgG BCR sequences with 2 mutations, we found that the seronegative group had higher SHM at baseline and week 3 (Figure 3B)”. Please explain in the manuscript what this means, especially what the highest SHM in seronegative group at baseline among the four groups of data means, and give some possible explanation. Is it just a variance due to the small sample size of this study?

3. Figure 3A, please correct the formatting issue.

4. Figure 3B, please correct the formatting issue (inconsistent color).

5. Line  342-345, “SHM was higher in the 28 convergent clones when compared to the top 50 expanded clone group (T-test with Welch’s correction; p<0.0001) and the remaining other clones in the repertoire (T-test with Welch’s correction; p<0.0001) (Figure 5B)”. I’m wondering how the dataset looks like if this piece of data is incorporated into Figure 4C, basically deviding the SHM in the 28 convergent clones into seropositive group and seronegative group.

6. Please discuss if Clone #8269 has any known functions beyond whether neutralizing SARS-CoV-2.

Author Response

We thank the reviewers for their time and consideration in reviewing our manuscript.  With this revision, we have addressed each of their comments and suggestions. Below is a point-by-point response to each reviewer comment. We believe this has greatly improved our study resulting in a stronger manuscript with increased clarity and scientific precision.

Reviewers' Comments:

Reviewer #3:

  1. It seems that all of the donors only got one dose of vaccination by week 3. If this is the case, please explicitly state it in the method section.

 RESPONSE: We thank the reviewer for this comment and have updated the methods to include this information.

“We enrolled health care workers from our children’s hospital with no known history of SARS-CoV-2 infection (n=4, seronegative) and with previous PCR-confirmed SARS-CoV-2 infection, 30-60 days prior to this study (n=5, seropositive). Peripheral blood was collected prior to vaccination with Pfizer mRNA vaccine (Comirnaty®, week 0) and after primary immunization (week 3). All participants received only one dose of vaccine before BCR analyses.”

  1. Line 235-237, “Within IgG BCR sequences with ≥2 mutations, we found that the seronegative group had higher SHM at baseline and week 3 (Figure 3B)”. Please explain in the manuscript what this means, especially what the highest SHM in seronegative group at baseline among the four groups of data means and give some possible explanation. Is it just a variance due to the small sample size of this study?

RESPONSE: We thank the reviewer for this question. We have elaborated on this point in the discussion. The low SHM for the seropositive individuals at baseline/week 0 could be due to a suppressed SHM rate which has been observed following COVID-19 infection recovery.

“Overall, the BCR repertoire after a single SARS-CoV-2 vaccination at week 3 did not differ greatly based on prior SARS-CoV-2 infection before immunization. At week 0 (baseline), the seronegative group had higher SHM compared to the seropositive group, this could be due to a decreased SHM rate in seropositive group, which has been observed in recently COVID-19 recovered individuals (15).  To a small degree, SHM increased in the seropositive group and decreased or remained the same in the seronegative group. It has been previously shown that SHM does not change in response to SARS-CoV-2 vaccination (14). Here, we observed a small decrease (IgG and IgM) or no change (IgA) for the seronegative group, which is in line with these previous findings. The increase in SHM observed in the seropositive group could indicate a reactivation and expansion of memory B cells which themselves have high SHM(15). Increased SHM in the BCR in response to vaccination after a previous infection has previously been shown for influenza (40). Additionally, when looking at SARS-CoV-2 recovered individuals after immunization, IgG+ memory B cells significantly increased after the first vaccine dose (41).”

  1. Figure 3A, please correct the formatting issue.

RESPONSE: We have changed the format of Figure 3A to show week 0 and week 3 for seropositive(red) and then week 0 and week 3 for seronegative (blue) separated by <2 mutations (left), and ≥2 mutations (right). This has also been corrected in the corresponding Supplemental Figure 1C and 2C to follow this same format.

  1. Figure 3B, please correct the formatting issue (inconsistent color).

RESPONSE: We have updated figure 3B, including making the lines around the violin plots thinner so that the colors do not appear different.

  1. Line  342-345, “SHM was higher in the 28 convergent clones when compared to the top 50 expanded clone group (T-test with Welch’s correction; p<0.0001) and the remaining other clones in the repertoire (T-test with Welch’s correction; p<0.0001) (Figure 5B)”. I’m wondering how the dataset looks like if this piece of data is incorporated into Figure 4C, basically deviding the SHM in the 28 convergent clones into seropositive group and seronegative group.

RESPONSE: This is a useful suggestion and was not something we had examined previously. We re-analyzed the data we found that the SHM is not different based on serotype within the 28 convergent clones. We found that the mean SHM value for seropositive convergent clones was 7.617 and mean SHM for seronegative convergent clones was 7.500. These groups were not statistically different from one another (T-test with Welch’s correction; p=0.8975). Due to the lack of a difference between the serotypes, and for clarity we have reverted to the original figure in the manuscript and we have updated the text to reflect this analysis.

“We then characterized the properties of the 28 convergent clones that were shared across all 9 individuals: HCDR3 length, V gene usage, and SHM. No difference was observed between the 28 convergent seropositive (mean =7.617) and seronegative groups (mean=7.500) (T-test with Welch’s correction p=0.8975). SHM was higher in the 28 convergent clone groups when compared to the top 50 expanded clone group (T-test with Welch’s correction; p<0.0001) and the remaining other clones in the repertoire (T-test with Welch’s correction; p<0.0001) (Figure 5B). The 28 convergent clone group had the highest SHM of any clone group observed in this study (convergent 28: 7.4027%; top 50: 3.813%; other clones:1.515%).”

  1. Please discuss if Clone#8269 as any known functions beyond whether neutralizing SARS-CoV-2.

RESPONSE: Clone #8269 does not have extensive functional research possibly due to its lack of neutralizing activities. This clone does bind SARS-CoV-1, binds spike protein, and has been isolated in multiple individuals in response to SARS-CoV-2 and SARS-CoV-2 vaccination. This does not rule out the possibility that it may have other important roles in mediating SARS-CoV-2 immunity, and those would be of interest to follow up on. The text has been updated to reflect this suggestion from the reviewer.

“Clone #8269 matched to a wide variety of targets including 100% alignment to: XG001, R121-3G10, R410-3D10, PDI-38, COVA1-27, H712443+K711941, R259-1F4, CV10, H712427+K71111927, and Shiakolas_53181-5. This clone used V4-59, had a HCDR3 length of 6, and a SHM of 0.759% (Table 1). These aligned antibodies were not reported to have neutralizing activities (28-33). Clone #8269 also had 83.33% alignment with XG002, R616-1E6, PDI-124, R849-3B4, R849-1F9, PDI-211 (Table 1). These antibodies also did not have known neutralizing activities (28-30). Clone #8269 has some cross-reactivity with SARS-CoV-1, binds the spike protein, and has been isolated from multiple human SARS-CoV-2 patients and SARS-CoV-2 vaccinees (29,34,35).  The short HCDR3 length of 6 amino acids could play a role in the alignment to multiple BCR sequences. These results identified three public clonotypes that were previously reported as SARS-CoV-2-specific in other studies, with two that had SARS-CoV-2 neutralization as a previously described function.”

“Highly convergent clones were found in our data set in response to vaccination. These clones were present across both serotypes, and 28 of these clones appeared in every individual in this study. This suggests similar responses to vaccination across serotypes. Early clonal convergence in response to SARS-CoV-2 is a feature that has been previously described in other datasets [18, 45]. These 28 convergent clones were unique as a group and had high levels of SHM and shorter HCDR3 lengths when compared to expanded and “other” clone groups, respectively. Use of the publicly available COVID antibody database revealed that three clones from our entire expanded clone set aligned to targets with greater than or equal to 80% homology of characterized SARS-CoV-2-specific B cells. Two of these public clonotypes (clone #34727, #13327) were previously found to generate SARS-CoV-2 neutralizing antibodies. The other public clone (clone #8269) had cross-reactivity with SARS-CoV-1 spike protein but was non-neutralizing. Functional studies of clone #8269 would be of future interest. These findings indicated that public clonotypes were generated in response to SARS-CoV-2 vaccination.”

Round 2

Reviewer 1 Report

The authors met all the points I have raised in the review thus I find it suitable for publication.

Reviewer 2 Report

The revision addressed my concerns and is good to go.